# The Regulatory Effects of a Formulation of *Cinnamomum osmophloeum* Kaneh and *Taiwanofungus camphoratus* on Metabolic Syndrome and the Gut Microbiome

**DOI:** 10.3390/plants9030383

**Published:** 2020-03-20

**Authors:** Ya-Yun Wang, Yu-Hsin Hsieh, K. J. Senthil Kumar, Han-Wen Hsieh, Chin-Chung Lin, Sheng-Yang Wang

**Affiliations:** 1Department of Forestry, National Chung-Hsing University, Taichung 402, Taiwan; wangyayun1015@gmail.com (Y.-Y.W.); mineral18@gmail.com (Y.-H.H.); charaljana14@rediffmail.com (K.J.S.K.); 2Taiwan Leader Biotech Company, Taipei 103, Taiwan; ck_hsieh@twleaderlife.com (H.-W.H.); johnson@twleaderlife.com (C.-C.L.); 3Agricultural Biotechnology Research Center, Academia Sinica, Taipei 128, Taiwan

**Keywords:** metabolic syndrome (MetS), *Cinnamomum osmophloeum* (CO), *Taiwanofungus camphoratus* (TC), high-fat diet (HFD), hyperglycemia, gut microbiota

## Abstract

The number of people with metabolic syndrome (MetS) is increasing year by year, and MetS is associated with gut microbiota dysbiosis. The demand for health supplements to treat or prevent MetS is also growing. *Cinnamomum osmophloeum* Kaneh (CO) and *Taiwanofungus camphoratus* (TC) are endemic to Taiwan. Both have been shown to improve the symptoms of MetS, such as dyslipidemia and hyperglycemia. Herein, we investigated the effect of CO, TC and their formulations on diet-induced obese mice. Male C57BL/6J mice were fed with a high-fat diet (HFD) for 10 weeks to induce MetS. After that, the mice were fed with HFD supplemented with CO, TC, and various CO/TC formulations, respectively, for 14 weeks. The changes in physiological parameters and the composition of the gut microbiome were investigated. The results indicated that CO, TC, and their formulations effectively reduced hyperglycemia, and tended to alleviate MetS in obese mice. Moreover, we also observed that CO, TC, and their formulations improved gut microbiota dysbiosis by decreasing the Firmicutes-to-Bacteroidetes ratio and increasing the abundance of *Akkermansia* spp. Our results revealed that CO and TC might have potential for use as a prebiotic dietary supplement to ameliorate obesity-related metabolic disorders and gut dysbiosis.

## 1. Introduction

Metabolic syndrome (MetS) is characterized by at least three of five risk factors, which are obesity, hypertension, dyslipidemia, insulin resistance, and hyperglycemia. Nowadays MetS is a global epidemic which increases the risk of developing type 2 diabetes, cardiovascular disease, heart disease, and stroke [1,2]. The occurrence of MetS is associated with the onset of obesity and type 2 diabetes. Taking the United States as an example, about 30.2 million adults had type 2 diabetes in 2017; around 1/4 (23.8%) of which are not aware that they have diabetes. Incidence of type 2 diabetes increases with age, reaching a high of 25.2% among US senior citizens (above 65 years old). Occurrence of prediabetes or MetS is about three times more [3]. It is recognized that MetS has become the major health hazard in the modern world.

Studies conducted on both human and animals have revealed that gut microbiota play a crucial role in the pathogenesis of MetS. Gut microbiota are the prime regulator of diet and the development of MetS. Dysbiosis, also known as dysbacteriosis strongly associated with several metabolic diseases, including diabetes, non-alcoholic fatty liver disease and chronic inflammatory diseases through modulating nutrients absorbance and energy intake from the foods, cholesterol metabolism, glucose metabolism and insulin resistance [2,4,5,6]. Recent studies have been demonstrated that beneficial gut microbiome, such as *Lactobacillus* [5], *Akkermansia muciniphila* [1] and *Bifidobacterium* [7] prevents and retard the development of MetS. Conversely, the pathogenic gut microbiome, including *Erysipelotrichaceae* [8], *Coriobacteriaceae* [9], and *Streptococcaceae* [10] are consistently associated with the development of MetS, such as diabetes, obesity, systemic inflammation and cardiovascular diseases. Therefore, regulating the high-fat diet (HFD)-induced dysbiosis of gut microbiome might be a novel therapeutic strategy for the prevention of MetS and its related diseases.

*Cinnamomun osmophloeum* Kaneh (CO) is a tree that is endemic to Taiwan. Its most distinguishing feature is that the composition of the leaf essential oil is similar to that of cinnamon. The leaves of CO are much sweeter and comparatively more aromatic than its related species growing in the region. Hussain and colleagues analyzed the leaf composition. Hussain et al. [11] reported that the methanolic extract of the leaves of CO is the sweetest substance; and it was found that trans-cinnamaldehyde (1.03% *w/w*) is responsible for this characteristics and it was estimated that trans-cinnamaldehyde is approximately 50-times sweeter than that of 0.5% *w/v* of sucrose solution. Accumulating evidences suggest that CO leaf essential oils possessed various bioactivities, including anti-bacterial [12], anti-termites [13], anti-mildew [14], and anti-fungal [15]. In addition to anti-microorganism studies, the potential application of CO leaves in food supplements is an interesting subject. It was found that CO leaf essential oils and its major compound, cinnamaldehyde possessed xanthine oxidase inhibitory and anti-hyperuricemia activities in mice [16]. In addition to essential oils, oral administration of aqueous leaf extracts of CO reduced total cholesterol, triglyceride and low-density lipoprotein cholesterol levels in hyperlipidemic hamsters [17].

*Taiwanofungus camphoratus* (TC; Syn. *Antrodia cinnamomea; Antrodia camphorate*) is also a unique and endemic medicinal fungus native to Taiwan. Traditionally, TC has long been used to treat various human illness, including food and drug intoxication, liver diseases, abdominal pain, diarrhea, hypertension, skin diseases, inflammation and even though for tumorigenic diseases. Recent scientific investigations revealed that the pharmacological effects of TC was go far beyond its original usage, as evidenced by that TC exhibited anti-oxidant, anti-inflammation, anti-microbial, anti-cancer, anti-aging, immunomodulatory, hepatoprotective and neuroprotective effects in vitro and in vivo models [18,19,20,21,22,23,24,25].

Since CO and TC possess diverse biological activities, they might also regulate metabolic syndrome. Furthermore, whether these two materials that are widely sold in the health food market have a synergistic effect was worth evaluating. Moreover, so far, only TC has been studied for its effect on the gut microbiota of animals. Therefore, in the present study, the metabolic syndrome regulation activity of CO, TC, their formulation and their association with modulation of the gut microbiota was evaluated by using an HFD-induced mouse model. 

## 2. Results

### 2.1. CO/TC Formulations Attenuated Symptoms of HFD-induced Metabolic Syndrome

To evaluate the effects of CO/TC formulations on MetS induced by HFD, male C57BL/six mice (n = six per group) were administrated water (vehicle control), Metformin (clinical drug), or CO/TC formulations for 14 weeks after 10 weeks of HFD induction. As shown in Figure 1a–e, compared with the normal diet (ND) group, mice that received HFD had significantly increased body weight (3.23-fold heavier than ND), epididymal adipose tissue weight (2.67-fold heavier than ND), and mild/moderate inflammation. However, treatment with metformin (Met) significantly reduced HFD-mediated body weight gain (3.18-fold less than HFD), epididymal adipose tissue inflammation. Among the CO/TC individual or formulation treatment groups, the CO treatment (only intake of CO leaves) significantly alleviated HFD-mediated body weight gain (1.09-fold less than HFD) (Figure 1b). Although there was no significant difference in epididymal adipose tissue weight (Figure 1c), most CO/TC treatments (except High-dose) tended to reduce the inflammation of adipose tissue, but not statistically significant (Figure 1d,e). To further examine the inhibitory effect of CO/TC treatments, the expression levels of the main pro-inflammatory factors related to metabolic diseases in adipose tissue also known as adipokines, including TNF-α, IL-1β and IL-6, were determined by qRT-PCR. Compared with the ND group, the mRNA expression levels of TNF-α, IL-1β and IL-6 markedly increased in the HFD group, whereas treatment with metformin significantly inhibited the HFD-induced inflammatory factors (Figure 1f–h). The data also showed that High-dose group displayed pronounceable activity in reducing the expression levels of these genes under HFD condition (Figure 1f–h). In particular, compared to the HFD group, all CO/TC treatments significantly down-regulated the mRNA expression level of IL-6 (Figure 1h). In addition, it is well known that nonalcoholic fatty liver disease is strongly associated with MetS [26]. Consequently, the effects of CO/TC formulations on hepatic steatosis were examined. Liver weight of the HFD group increased 1.48-fold over the ND group (Figure 2a), and symptoms of hepatic lipid deposition were found in the HFD group (Figure 2b–d). As expected, metformin treatment improved liver pathological changes induced by a high-fat diet in mice (Figure 2a–d). CO/TC formulations reduced liver weight in high-fat diet induction, although the effects did not reach statistical significance (Figure 2a). Moreover, the CO/TC formulation was unable to improve fatty change in liver, but decreased generation of macro-vesicular steatosis (Figure 2b–d).

### 2.2. CO/TC Formulations Mitigated HFD-induced Dysglycemia

It is well known that high cholesterol and dysglycemia are associated with obesity. After HFD treatment, the levels of total cholesterol (2.5-fold higher than ND) and fasting glucose (1.59-fold higher than ND) were significantly elevated, and glucose tolerance was also impaired (Figure 3a–d). However, CO/TC formulations did not significantly decrease the level of total cholesterol, a result mirrored in the Metformin group (Figure 3a). Notably, the levels of fasting glucose in each of the CO/TC formula groups were significantly reduced; the CO, Medium-dose and High-dose groups even performed nearly as well as similar to the Metformin group (Figure 3b). In oral glucose tolerance tests, the Medium-dose group exhibited a similar effect to the Metformin group, both significantly improved glucose tolerance in HFD mice. Other test groups also had the same tendency to restore impaired glucose tolerance in obese mice, although the data were not statistically significant to the HFD group (Figure 3c,d). Furthermore, there was no significant difference in the levels of BUN and creatinine between mice that received the CO/TC formulation treatment and ND (Figure 3e,f), indicating that CO/TC formulas had no effects on kidney function.

### 2.3. CO/TC Formulations Altered Gut Microbiota Composition in HFD-fed Mice

An increasing number of studies have confirmed that the development of MetS and related diseases is strongly associated with the gut microbiota [4,27]. Therefore, the effect of CO/TC formulations on gut bacteria was analyzed by 16S rRNA sequencing. α-Diversity of the samples was measured by observed OTUs and the Shannon diversity index. Observed OTU results represented species richness and the Shannon index indicated diversity; the higher the number recorded, the higher the species richness and diversity in the samples. The results revealed that both the OTUs and Shannon indices of HFD group were significantly decreased by 20.92% and 11.36%, respectively, compared with the ND group (Figure 4a,b). However, Metformin and all CO/TC treatments groups were unable to eliminate this effect, Metformin even exacerbated this phenomenon (OTUs: 52.42% lower than ND, Shannon: 41.97% lower than ND) (Figure 4a,b). Principal co-ordinate analysis (PCoA) of the weighted UniFrac distance metric illustrated that the bacterial composition in mice cecal contents clustered according to host diet (high fat diet and normal diet), but was not clearly separated by CO/TC treatments (Figure 4c). As shown in Figure 4d, the phylum of *Firmicutes* and *Bacteroidetes* accounted for at least 65% in the taxa, especially in the ND group, where they represented up to 85%. After the calculation, the relative abundance of *Firmicutes* to *Bacteroidetes* (F/B ratio) in the HFD group was elevated compared to that of the ND group and High-dose treatment significantly improved this phenomenon (Figure 4e). Importantly, the relative abundance of *Akkermansia* in mice was increased after receiving the CO/TC formulations (Figure 4f), which correlated with an improved metabolic profile. In addition, the CO/TC formulations also decreased the level of *Mucispirillum* and *Blautia* compared with the HFD group (Figure 4g,h). Both bacteria are linked to obesity and diabetes. However, the clinical drug, Metformin, displayed better enhancement of the F/B ratio and an abundance of *Akkermansia* (Figure 4e,f), and much less presence of *Mucispirillum* and *Blautia* (Figure 4g,h).

## 3. Discussion

Previous studies have shown that TC can reduce obesity and attenuate hyperglycemia in rodents, and may act by regulating the composition of the gut microbiota [28,29,30]. It has also been confirmed that CO inhibits hyperlipidemia and hyperglycemia [17,31]. However, whether a CO/TC formulation could enhance MetS regulation activity or not remained unclear. In this study, we used a long-term diet-induced obese mouse model to evaluate the efficacy of CO/TC formulations on regulating MetS and their impact on the composition of gut microbiota. We demonstrated that CO/TC formulation significantly suppressed the level of fasting plasma glucose in obese mice and increased the proportion of beneficial microbiota in the intestines. However, our results did not show a synergistic effect for the CO and TC formulation.

It is no surprise that our results are similar to previous studies [17,28,29,30,31] showing that both CO and TC efficiently controlled impaired glucose metabolism and improved hyperglycemia. However, our results are inconsistent with previous studies in that TC did not inhibit weight gain or reduce lipid accumulation [28,29,30]. This might be due to the longer-term high-fat diet induction experiment (HFD for 24 weeks) in our study. This is much longer than other studies (HFD for 12 weeks) [28,29]. Furthermore, our experimental design made mice obese (HFD induction for ten weeks) before starting the medication. Thus, we speculate that TC may not have long-term effectiveness in reducing obesity, and can only prevent, but not treat obesity.

It has been demonstrated that the increased number of adipose tissue mononuclear cells, namely the pro-inflammatory macrophages are directly proportional to the production of pro-inflammatory cytokines, also known as adipokines, a phenomenon that contributes to the development of insulin resistance and type 2 diabetes in obese animals [32]. Increased inflammation and the expression of higher pro-inflammatory cytokines can be found in the adipose tissues of HFD animals, which are associated with an increased risk of developing MetS-related complications [33]. In this study, we observed mild/moderate inflammation in HFD fed mice, which was significantly inhibited by metformin. All the treatments of CO/TC reduced inflammation in adipose tissues; however, the inhibitory effect was not statistically significant. Since the TC and CO were potent anti-inflammatory agents, here we observed that a high dose of CO/TC formulation reduced the mRNA expression of TNF-α and IL-1β, while all CO/TC treatments groups significantly decreased the mRNA expression of IL-6. In particular, the inhibitory effect of IL-6 mRNA by a high dose of formulation was highly comparable with metformin. These results partially explained the anti-inflammatory effect and improvement of dysglycemia in the obese mice.

Obesity is usually associated with hepatic steatosis, a condition of ectopic fat deposition in the liver. Conventionally, it is believed that hepatic steatosis is primarily triggered by imbalanced lipogenesis and lipid peroxidation, while recent studies postulate that the metabolic communication between the liver and adipose tissues may play a vital role for the development of hepatic steatosis in the context of obesity [34]. Indeed, adipose inflammation induced by pro-inflammatory cytokines or adipokines greatly exacerbated hepatic steatosis in obesity, and suppressing this inflammation can completely block HFD-induced hepatic steatosis. In this study, we found that an HFD caused an increase in weight and the fatty change of micro- and macro-vesicles. Treatment with CO/TC inhibited micro- and macro-vesicles, which correlated with reduced liver weight. However, these inhibitory effects were not statistically significant. It can be speculated that the anti-inflammatory properties of CO/TC may be a compensatory mechanism preventing HFD-induced hepatic steatosis.

Adipose inflammation and ectopic lipid deposition are the primary causes of diabetic manifestation in obesity [34]. Recent studies have shown that adipose tissue-associated pro-inflammatory cytokines causatively contribute to insulin resistance. Indeed, inhibition of adipose inflammation by blocking IL-1 improved hyperglycemia and hyperlipidemia [35,36]. In this study, the HFD-fed mice exhibited an apparent increase in fasting glucose and total cholesterol levels in the circulation. The increased levels of fasting glucose were significantly inhibited by all the CO/TC treatments, whereas neither CO/TC treatment nor metformin failed to inhibit the HFD-mediated elevation of total cholesterol levels, which is in contradiction to previous studies that asserted that TC and CO inhibited total cholesterol levels in obese mice [28,37].

It is worth noting that our results demonstrated for the first time that CO and CO/TC formulations changed the composition of gut microbiota, and reduced the F/B ratio in HFD mice. Many studies have proved that the F/B ratio is associated with obesity and metabolic syndrome [38,39,40]. *Firmicutes* gradually increased while *Bacteroidetes* decreased with increasing obesity, i.e., the F/B ratio rose. Conversely, a lower F/B ratio was observed in leaner animals. This result is consistent with previous results found in mushroom-derived prebiotics treatments [28,41]. Furthermore, the relative abundance of *Akkermansia* was increased in every CO/TC treatment group. Recent evidence indicates that some specific bacteria affect the physiology and metabolism of the host and then alter the development of metabolic disorders. For example, *Akkermansia* spp. and *Bacteroides fragilis* play an essential role in the therapeutic efficacy of Metformin [42,43]. Among the *Akkermansia* spp., *Akkermansia muciniphila* is known to have numerous health benefits [44]. Everard and colleagues showed that *A. muciniphila* treatment reversed metabolic disorders and restored colon mucosal barrier dysfunction in HFD mice [45]. Moreover, Dao’s group found that *A. muciniphila* abundance is associated with blood glucose stability and metabolic health in adults [46]. Furthermore, enriching *Akkermansia* spp. through diet or drug supplementation usually improves metabolic parameters [44,47]. This means that CO/TC may increase the abundance of *Akkermansia* spp. to exert a beneficial effect. Our study also found that the relative abundance of *Mucispirillum* spp. and *Blautia* spp. were increased in obese mice and decreased by CO/TC administration. Previous studies have shown that circulating leptin concentrations are positively correlated with the abundance of *Mucispirillum* spp. and showed the highest relative abundance in obese mice [48]. *Blautia* spp. increases in the overweight phenotype [49]. Moreover, high abundance of *Blautia* is associated with glucose intolerance and the presence of *Blautia* is positively correlated with Type 2 diabetes [50,51]. Our data were consistent with this result and indicated that CO/TC might suppress *Mucispirillum and Blautia* colonization in the intestine which may have a beneficial effect. Taken together, these results suggest that CO/TC and their formulations may ameliorate HFD-induced MetS by regulating the dysbiosis of gut microbiota in multiple ways.

## 4. Materials and Methods 

### 4.1. Preparation of CO and TC Extracts

The leaves of *Cinnamomum osmophloeum* (CO) were collected in September 2015 from Yunlin County, Taiwan. The species were identified by Professor Sheng-Yang Wang, and the voucher specimens were deposited at the Herbarium of the Department of Forestry, National Chung-Hsing University (Voucher No: T. C. 3). The hot water extract of CO leaves was prepared by the following method: 1 kg of CO leaves were air-dried, powdered and extracted by boiling with 15 L hot distilled water for 30 min. Then the filtrates were collected and concentrated by lyophilization, which yields 165 g of CO hot water extract (COE; yield = 16.5%, w/dry weight of leaves). Solid-state cultured *Taiwanofungus camphoratus* (TC) was provided by the R&D Center of Taiwan Leader Biotechnology Corp., Taiwan. Fresh and air-dried mycelia of TC (400 g) were soaked in ethanol (4 L) for 5 days, then the extract was concentrated under a vacuum to yield the ethanol extract (TCE; 50 g).

### 4.2. Animals and Experimental Design

Animal experiments were designed in accordance with the *Guidebook for the Care and Use of Laboratory Animals* of the Chinese-Taipei Society of Laboratory Animal Science and approved by the same society (No. 106-007^R^). A mouse model of diet-induced obesity was used in this study. Three-week-old male C57BL/6J mice purchased from BioLASCO Taipei, Taiwan Co., Ltd. and were housed in individual plastic cages under controlled light conditions (12 h light-dark cycle), with ad libitum access to food and water. After acclimatization for a week, the mice were randomly assigned to eight groups (6 mice per group). One group was fed with normal-chow diet (ND, 13.38% of energy from fat; LabDiet 5001; LabDiet, St Louis, MO, USA) and the other groups were fed with a high-fat diet (HFD, 60% of energy from fat; TestDiet 58Y1; TestDiet, St Louis, MO, USA) for 10 weeks. Then, the HFD-fed groups were divided into HFD with administration of water as vehicle control, HFD plus daily administration of 250 mg/kg/day metformin as a clinical drug control (Met group), HFD plus daily administration of 230 mg/kg/day TCE (TC group), HFD plus daily administration of 500 mg/kg/day COE (CO group), HFD plus daily administration of 230 mg/kg/day TCE and 200 mg/kg/day COE (Low-dose group), HFD plus daily administration of 230 mg/kg/day TCE and 500 mg/kg/day COE (Medium-dose group), HFD plus daily administration of 230 mg/kg/day TCE and 1000 mg/kg/day COE (High-dose group) by gavage for 14 weeks. The total experimental period was 24 weeks. During the experimental period, food intake and body weight were recorded and fasting blood glucose was measured weekly. At the end of the experiment, doing oral glucose tolerance test and the organs, tissues, cecal feces, and serum samples of each mouse were collected after euthanasia (CO_2_ inhalation) for subsequent analyses.

### 4.3. Histology Observation

The epididymal fat and liver were fixed in 10% formalin for 1 week. After tissue pruning and paraffin embedding, 2 mm thick sections were made and stained with hematoxylin and eosin solution. Then, the sections were observed by optical microscopy and the histological score was measured.

### 4.4. Biochemical Analysis

The plasma levels of blood urea nitrogen (BUN), creatinine, and total cholesterol levels were measured spectrometerically. All analyses were performed by using Express Plus Automatic Clinical Chemistry Analyzer with the manufacturer’s instructions (Chiron Diagnostics Corporation, Oberlin, OH, USA).

### 4.5. Oral Glucose Tolerance Tests

Glucose tolerance tests were executed after overnight fasting. Blood glucose were measured by a glucometer (GT-1650, Arkray, Japan), and mice blood samples were taken from the tail vein at 0, 30, 60, 120, and 180 min after oral gavage (1 g/kg body weight).

### 4.6. RNA Preparation and qPCR Analysis

The total RNA of epididymal adipose tissue was extracted using a Total RNA Miniprep Purification Kit (GeneMark, Taipei, Taiwan). DNA was removed with a TURBO DNA-free™ Kit (Invitrogen, Waltham, MA, USA) using the routine DNase treatment protocol. Purified RNA was converted to cDNA immediately by using SuperScript^®^ IV Reverse Transcriptase as indicated in the SuperScript™ IV First-Strand Synthesis System (Invitrogen, Waltham, MA, USA). qPCR reactions were performed with equal volumes of cDNA, forward and reverse primers, PowerUp™ SYBR™ Green master mix (Applied Biosystems, Foster City, CA, USA) on Applied Biosystems detection instruments and software. Threshold cycle (Ct) values were automatically generated by the 7500 Real-Time PCR software. Lack of variation in PCR products and the absence of primer–dimers were ascertained from the melt curve profile of the PCR products. Relative quantification was analyzed by the 2^−ΔΔCt^ method. Glyceraldehyde-3-phosphate dehydrogenase (GAPDH) was chosen as housekeeping gene. The primer sequences of each gene for qPCR were as follows. TNF-α: forward primer (F), 5′-TAGCCAGGAGGGAGAACAGA-3′; reverse primer (R), 5′-TTTTCTGGAGGGAGATGTGG-3′; IL-1β: forward primer (F), 5′-TTGAAGAAGAGCCCATCCTC-3′; reverse primer (R), 5′-CAGCTCATATGGGTCCGAC-3′; IL-6: forward primer (F), 5′-CCGGAGAGGAGACTTCAC-3′; reverse primer (R), 5′-TCCACGATTTCCCAGAGA-3′; GAPDH: forward primer (F), 5′-TCAACGGCACAGTCAAGG-3′; reverse (R), 5′-ACTCCACGACATACTCAGC-3′.

### 4.7. Gut Microbiota Analysis

Cecal feces were snap-frozen in liquid nitrogen and stored at −80 °C. Bacterial DNA was extracted by QiAamp fast DNA stool Mini Kit (Qiagen, Hilden, Germany). The V3-V4 regions of 16S rRNA gene was amplified using the specific primers (341F-805R) containing a unique barcode to tag PCR products. All PCR reactions were carried out in 25 μL reactions with 0.5 μL of KAPA High-Fidelity PCR Master Mix (Kapa Biosystems, Wilmington, MA, USA), 0.5 μM of forward and reverse primers, and about 1 ng template DNA. Thermal cycling began with an initial denaturation at 95 °C for 3 min, followed by 30 cycles of denaturation at 95 °C for 30 s, annealing at 57 °C for 30 s, and elongation at 72 °C for 30 s; and finally, within 72 °C for 5 min. PCR products were mixed in equidensity ratios. Then, the mixture of PCR products were purified with QIAquick Gel Extraction Kit (Qiagen, Hilden, Germany). Sequencing libraries were generated using Truseq nano DNA Library Prep Kit (Illumina, San Diego, CA, USA) following the manufacturer’s recommendations and index codes were added. The library quality was assessed on the Qubit@ 2.0 Fluorometer (Thermo Scientific, Waltham, MA, USA) and Agilent Bioanalyzer 2100 system. Finally, the library was sequenced on an Illumina Miseq platform, generating 300 bp paired-end reads. All reads were clustered into Operational Taxonomic Units (OTUs) using Mothur v.1.39.5 with 97% identity. Alpha and beta diversity analysis were performed by using QIIME (Quantitative Insights into Microbial Ecology) software. The alpha diversity indices including observed OTUs and Shannon diversity were determined. Beta diversity was assessed by the weighted UniFrac distance and visualized using principal coordinates analysis (PCoA).

### 4.8. Statistical Analysis

All data were reported as mean ± SD and analyzed using GraphPad Prism 7 software for Windows (GraphPad Software, La Jolla, CA, USA). Non-parametric statistical comparisons applied one-way ANOVA with Tukey’s post-hoc tests.

## 5. Conclusions

In this work, we investigated the physiological effects of CO/TC formulations in diet-induced obese mice and their impact on gut microbiota. Our results demonstrated that CO/TC could improve abnormal blood glucose regulation and restore the balance of gut microbiota in a longer-term HFD-induced mouse model. Both CO and TC are therefore bona fide dietary supplements and health products. The optimized CO/TC formulation and its molecular mechanism still need further exploration.

## Figures and Tables

**Figure 1 plants-09-00383-f001:**
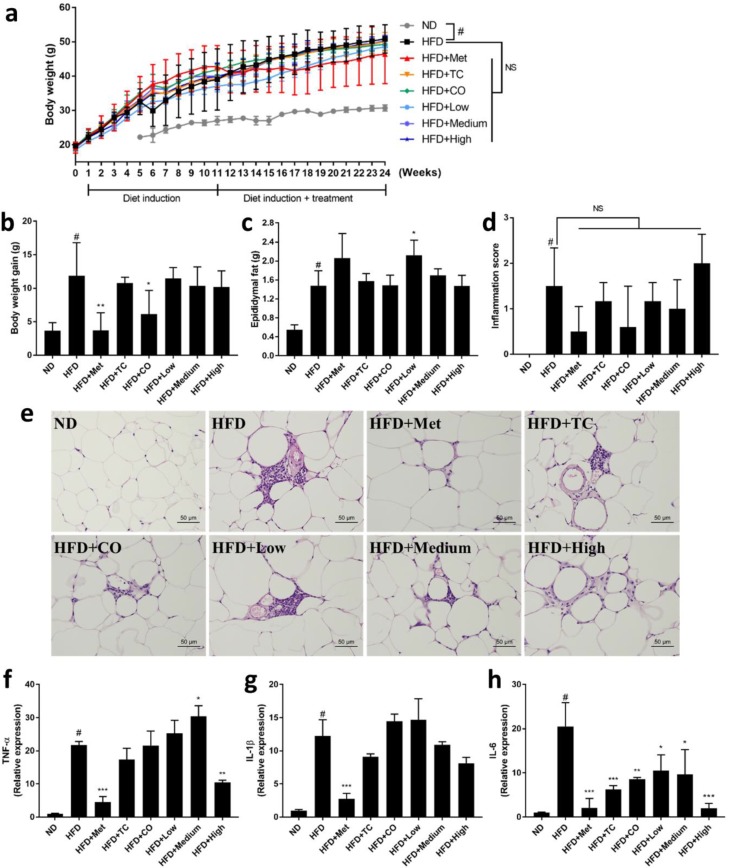
Effects of CO/TC formulations on (**a**) body weight, (**b**) body weight gain, (**c**) epididymal fat, (**d**) inflammatory index of adipose tissue, and (**e**) epididymal adipocyte microsections stained with H&E, and the relative expression of (**f**) TNF-α, (**g**) IL-1β, (**h**) IL-6 in epididymal adipose tissue. Values are means ± SD for n = 6, and analyzed by one-way ANOVA with Tukey post-hoc test. ^#^
*p* < 0.05 was considered significant for ND vs. HFD. *** *p* < 0.001, ** *p* < 0.01, * *p* < 0.05 vs. HFD group. NS, represents no significance difference (*p* > 0.05).

**Figure 2 plants-09-00383-f002:**
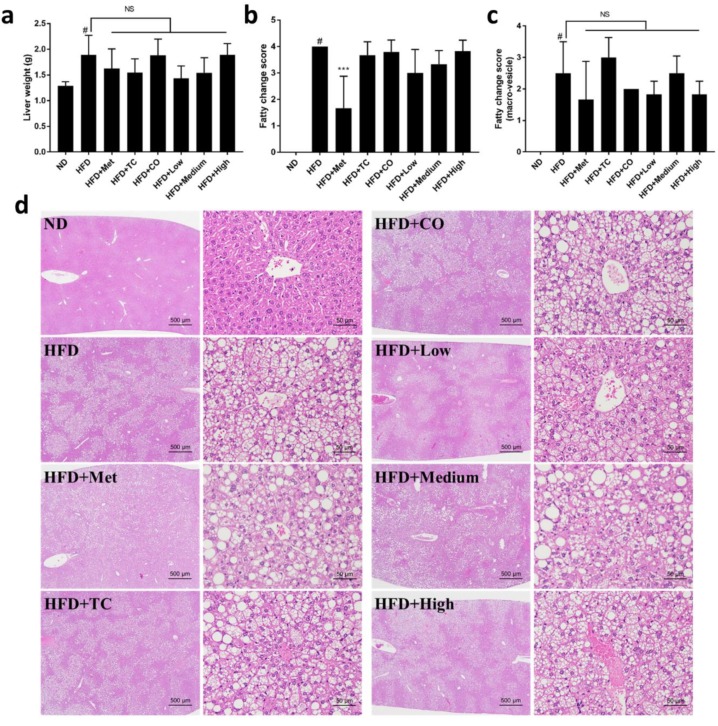
Effects of CO/TC formulations on hepatic steatosis (n = 6). (**a**) Liver weight. (**b**) Fatty change score of the livers. (**c**) Fatty change score with macro-vesicles of the livers. (**d**) Liver microsections stained with H&E. Data are reported as mean ± SD and analyzed by one-way ANOVA with Tukey post-hoc test. ^#^
*p* < 0.05 was considered significant for ND vs. HFD. *** *p* < 0.001 vs. HFD group. NS, represents no significance difference (*p* > 0.05).

**Figure 3 plants-09-00383-f003:**
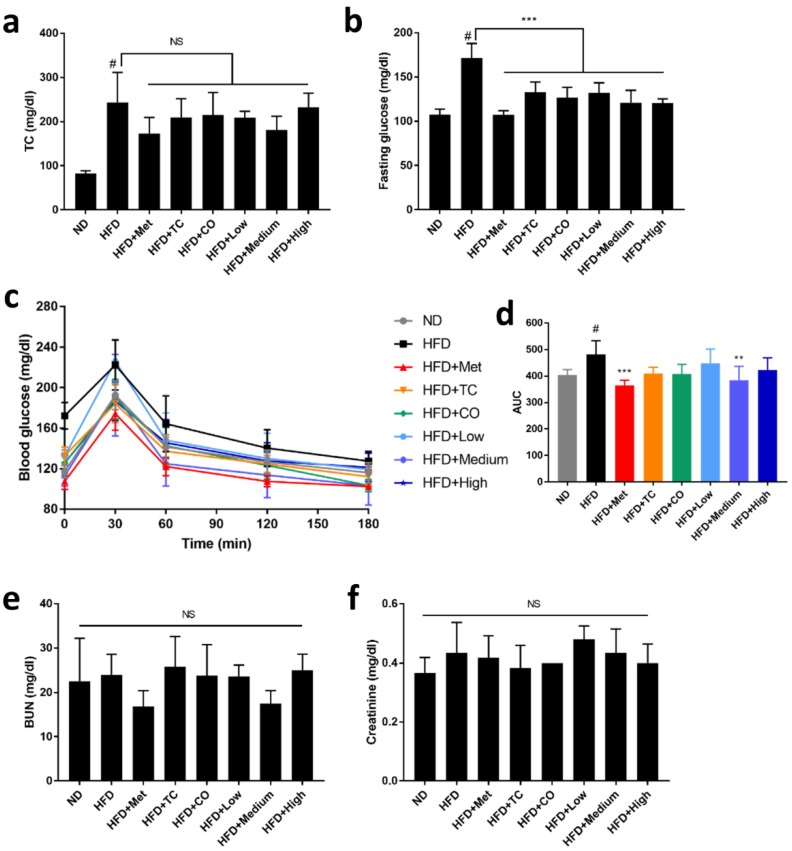
Effects of CO/TC formula on serum biochemical index (n = 6). (**a**) Total cholesterol. (**b**) Fasting glucose. (**c**) Oral glucose tolerance test and (**d**) its area under the curve (AUC). (**e**) Level of blood urea nitrogen. (**f**) Level of creatinine. Data are reported as mean ± SD and analyzed by one-way ANOVA with Tukey post-hoc test. ^#^
*p* < 0.05 was considered significant for ND vs. HFD. *** *p* < 0.001, ** *p* < 0.01 vs. HFD group. NS, represents no significance difference (*p* > 0.05).

**Figure 4 plants-09-00383-f004:**
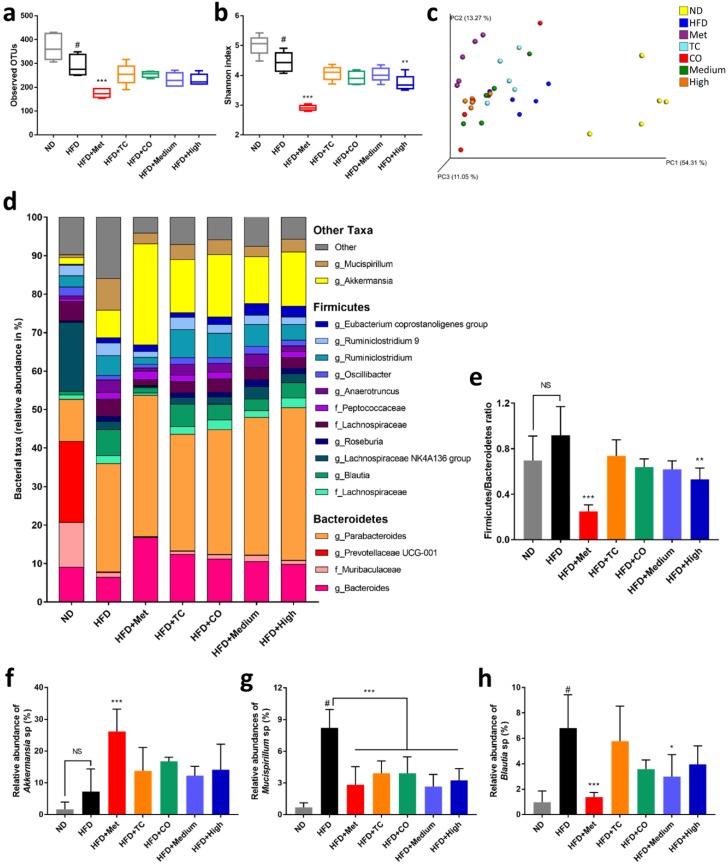
CO/TC formula alters gut microbiota composition in HFD-fed mice (n = 5). Alpha diversity assessed by (**a**) the Shannon index and (**b**) observed species. (**c**) Principal co-ordinate analysis (PCoA) of the weighted UniFrac metric for fecal microbiota. (**d**) Taxa summary of samples. OTUs were combined if their deepest taxonomic classifications were identical, and taxa with less than 1% abundance were combined into ‘Other’. (**e**) The ratio of *Firmicutes* to *Bacteroidetes*. (**f**) The relative abundances of genus *Akkermansia*. (**g**) The relative abundances of genus *Mucispirillum*. (**h**) The relative abundances of genus *Blautia*. Data are reported as mean±SD and analyzed by one-way ANOVA with Tukey post-hoc test. ^#^
*p* < 0.05 was considered significant for ND vs. HFD. *** *p* < 0.001, ** *p* < 0.01, * *p* < 0.05 vs. HFD group. NS, represents no significance difference (*p* > 0.05).

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
