# Peer review of "The Regulatory Effects of a Formulation of Cinnamomum osmophloeum Kaneh and Taiwanofungus camphoratus on Metabolic Syndrome and the Gut Microbiome"

_plants, 2020, doi:10.3390/plants9030383_

Round 1

Reviewer 1 Report

  It is judged that there is no problem in accepting this paper because the correction of the question has been made appropriately.

Reviewer 2 Report

No further comments and it is much improved now.

This manuscript is a resubmission of an earlier submission. The following is a list of the peer review reports and author responses from that submission.

Round 1

Reviewer 1 Report

The present study shows the regulatory effects of the formulation with Cinnamomum osmophloeum Kaneh and Taiwanofungus camphoratus on metabolic syndrome and gut microbiome. The design of study constructive and also the results is clearly showed.

For the publication, several point should be addressed.

According to abstract, CO/TC formulations feed for 14 weeks. (line 19). But, the results section, feeding period was 12 weeks. (line 90). Which one is correct? Please provide approval number for this experiment. “Physiological parameters---“, in line 20 should be “physiological parameters”

Reviewer 2 Report

Discussion and hypothesis of manuscript are very weak.The experimental design and the discussion of data are very poor. The set of analyses is scarce.

Reviewer 3 Report

Wang et al showed the positive outcomes of using Cinnamomum osmophloeum Kaneh and

Taiwanofungus camphoratus on metabolic syndrome and gut microbiome.

It’s a good study showing the properties of CO and TC formulations.

Manuscript has hundreds of grammatical errors throughout the paper.

comments:

Authors should check the effect of CO and TC effects on insulin tolerance test ‘’Among the CO/TC formulation treatments, CO group (only 97 intake the CO leaves) significantly alleviated body weight gain (93% lower than HFD) (Fig. 1b). ‘’ correct the % data…how is 93% possible??? Why only male mice were used Put scale bars to each of the sections in fig 1 and 2 Line 117, ‘’ It is well known that high cholesterol and dysglycemia are associate with obesity. After HFD 118 treatment, the levels of total cholesterol (196% higher than ND) and fasting glucose (59% higher than ND) were significantly elevated, and glucose tolerance was also impaired (Fig. 3a - d).’’ please correct 196%..... Line 94-96 ‘’ Besides epididymal adipose tissue weight, the Metformin treatment (Met) alleviated body weight 95 gain (218% lower than HFD), inflammation, liver weight (16% lower than HFD), and hepatic lipid 96 deposition comparing with HFD group’’ ….please correct the fold change....recalculate the data In fig legends 1, where is ‘’h’’ ???? correct it to ‘’e’’ in the fig legends Please check mRNA expression of IL-6, TNF-alpha, IL-1beta in adipose tissues and serum TNF-alpha by ELISA In Fig 2 upper panel, a, b, c, what are they comparing treatment groups with??? All the treatment groups should be compared with HFD group. This is same for fig 3. Please get your manuscript corrected by an expert before submitting the next time as it will lead to difficulty in understanding.